

# Quantum bit threads

### Andrew Rolph⋆

Institute for Theoretical Physics, University of Amsterdam,
1090 GL Amsterdam, The Netherlands
Martin A. Fisher School of Physics, Brandeis University,
Waltham, MA 02453, USA

⋆ andrew.d.rolph@googlemail.com

## Abstract

We give a bit thread prescription that is equivalent to the quantum extremal surface prescription for holographic entanglement entropy. Our proposal is inspired by considerations of bit threads in doubly holographic models, and equivalence is established by proving a generalisation of the Riemannian max-flow min-cut theorem. We explore our proposal's properties and discuss ways in which islands and spacetime are emergent phenomena from the quantum bit thread perspective.



# 1 Introduction

The connection between quantum information and geometry is one of the deepest aspects of quantum gravity to have emerged in recent decades, and this connection is manifest in the Ryu-Takayanagi (RT) formula which relates von Neumann entropies in holographic CFTs to minimal area surfaces in their bulk duals [1]:

$$S(A) = \frac{1}{4G_N} \min_{m \sim A} \text{Area}(m) \, . \tag{1}$$

Bit threads [2] are a flow-based reformulation of holographic entanglement entropy that was introduced in part to address some of the conceptual issues arising from the surface-based RT prescription. Bit threads replace the minimisation over boundary-anchored surfaces with maximisation of boundary flux of flows into the bulk. In the original 'classical' bit thread prescription, the von Neumann entropy of boundary region $A$ equals the maximum flux of a divergenceless, norm-bounded vector field $v^\mu(x)$ out of that boundary region:

$$S(A) = \max_v \int_A n_\mu v^\mu \, , \tag{2}$$

with $v$ subject to the constraints that

$$\nabla_\mu v^\mu = 0 \text{ and } |v| \leq \frac{1}{4G_N} \, . \tag{3}$$

Equivalence of this prescription to RT follows from a generalisation of the max flow-min cut theorem (MFMC) to continuous Riemannian manifolds [3]. The classical RT surface, as a minimal area surface, acts a bottleneck for the classical bit threads.

Both the RT formula and bit thread prescription (2) are only applicable to time symmetric states and have corrections at finite $N$ and coupling $\lambda$. At finite $\lambda$, when the effective bulk gravitational action has higher curvature terms, the RT prescription receives Wald-like corrections to the area functional [4–6], and these corrections are accounted for in the bit thread formulation with a spacetime-dependent norm bound [7]. To be accurate at finite $N$ the RT formula is modified to the quantum extremal surface (QES) prescription [8] which includes a bulk entanglement entropy contribution:

$$S(A) = \min_{m \sim A} \left( \frac{\text{Area}(m)}{4G_N} + S_{\text{bulk}}(\sigma(m)) \right) \, . \tag{4}$$

$\sigma(m)$ is the subregion of the bulk slice whose boundary is $\partial \sigma(m) = A \cup m$.

Quantum corrections to the bit thread prescription were briefly considered in the original paper [2], where the authors qualitatively suggested that the quantum correction be accounted for by allowing the bit threads (integral curves of $v$) to start and end at points in the bulk.[1] Loosely speaking the physical idea is to allow bit threads to tunnel between entangled bulk degrees of freedom, and this has a hope of capturing the bulk entanglement entropy term given in (4) because threads tunnelling across the RT surface allows for additional flux from $A$ roughly in proportion to the amount entanglement across that surface.

In this paper we find a quantum bit thread prescription: a flow-based prescription for calculating entanglement entropy in holographic CFTs that is accurate to all orders in $1/N$.[2]

---

[1]See also [9] for a discrete formulation of bit threads on MERA tensor networks with sources and sinks.

[2]Regarding bulk entanglement entropy, we neglect graviton fluctuations and potential issues of bulk Hilbert space factorisation, as in the FLM and QES proposals. Also, to focus on the generalisation to finite $N$, we neglect higher curvature corrections to the bulk gravitational action and assume that the state is time-reflection symmetric; a fully general prescription, which we leave for future work, would not make these simplifying assumptions.

This is in contrast to the original bit thread prescription, which is accurate only to leading order in $1/N$.

The prescription is

$$S(A) = \max_{v} \int_A n_\mu v^\mu \,, \tag{5}$$

with $v$ subject to the constraints

$$|v| \leq \frac{1}{4G_N} \,, \quad \text{and} \quad \forall (\sigma \in \Omega_A) : \quad \left( -\int_\sigma \nabla_\mu v^\mu(x) \leq S_{bulk}(\sigma) \right) \,, \tag{6}$$

where $\Omega_A$ is the set of all bulk homology regions for $A$:

$$\Omega_A := \{ \sigma \subseteq \Sigma : \partial \sigma \supseteq A \} \,. \tag{7}$$

A homology region $\sigma \in \Omega_A$ can be thought of as a time slice of a possible entanglement wedge for $A$; its boundary is $\partial \sigma = m \cup A$. The prescription has replaced minimisation over surfaces with maximisation over flows; there are no surfaces $m$ directly involved in the objective function or the constraints. We prove that (5) is equivalent to the QES prescription (4) using a technique from convex optimisation for mapping maximisation problems to equivalent minimisation problems and vice versa.

The key difference between the quantum and classical bit thread prescriptions is that the divergencelessness condition has been replaced. The new constraint allows threads to end at bulk points, but bounds the total number that can end in any given bulk homology region by the von Neumann entropy of the region. Bit threads in flux-maximising flows have the appearance of jumping between entangled bulk regions.

Quantum extremal surfaces and islands, which are not part of the bit thread prescription, appear as properties of flux-maximising flows. Quantum extremal surfaces are bottlenecks to the flow - just as RT surfaces are bottlenecks to bit threads in the classical prescription. Islands are bulk regions where so many bit threads reappear for a flux-maximising flow that the region's boundary is a novel disconnected bottleneck to the flow.

We also consider bit threads in doubly holographic models [10]. The benefit of this class of models is that the bulk entanglement entropy can be computed using the classical RT formula in one higher dimension, so in these models the quantum bit thread prescription in $\text{AdS}_{d+1}$ in a sense follows directly from the behaviour of classical bit threads along the boundary in the highest dimensional picture.

*Outline*

In section 2 we gain some intuition for quantum bit threads by considering classical bit threads in doubly holographic models, and we propose and comment on a quantum bit thread prescription. In section 3 we prove the equivalence of our quantum bit thread prescription to the quantum extremal surface prescription, using tools from convex optimisation. In section 4 we conclude with a discussion of the connection of quantum bit threads with ER=EPR, emergent spacetime, and islands, and talk about possible future directions to work towards. In appendix A we prove that quantum bit threads satisfy the nesting property of flows, which is sufficient for a flow-based holographic proof of strong subadditivity to all orders in $1/N$.

*Note*: As this work neared completion we learned of work by Cesar Agón and Juan Pedraza [11] which has partial overlap in scope, since they also study modifications of the bit thread proposal to account for bulk quantum corrections, and we have arranged to submit simultaneously to arXiv. Their prescription is only accurate to next-to-leading order in $1/N$, not to all orders, and also requires the position of the RT surface as input.

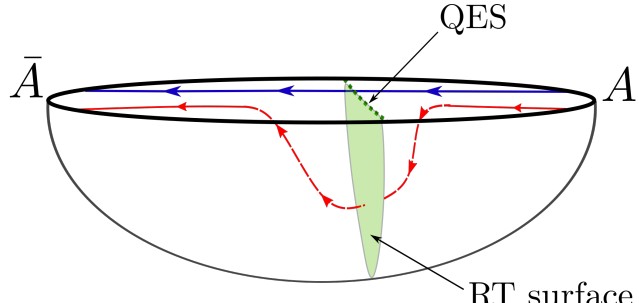

Figure 1: Bit threads on a time slice of a doubly holographic model. The geometry is asymptotically $AdS_{d+2}$ with an end-of-the-world brane. The bit threads are divergenceless; we assume the bulk is classical. For flux-maximising flows, some flow lines go from $A$ to $\bar{A}$ by moving through the $AdS_{d+1}$ boundary while others move into the bulk. An observer restricted to the $AdS_{d+1}$ boundary sees threads appearing and disappearing on either side of the QES (see figure 2).

## 2 Quantum bit threads

In this section we will give a flow-based bit thread prescription which is equivalent to the quantum extremal surface (QES) prescription [8]

$$S(A) = \min_{m \sim A} \left( \frac{\text{Area}(m)}{4G_N} + S_{\text{bulk}}(\sigma(m)) \right), \tag{8}$$

where $\sigma(m)$ is the subregion of the bulk slice whose boundary is $\partial \sigma(m) = A \cup m$. We assume time reflection symmetry to separate the challenges of creating covariant and quantum bit thread prescriptions.

The generalised entropy that features in the QES prescription has two sources of UV divergences: from the bulk entanglement entropy and counterterms in the gravitational action. We assume that these divergences cancel each other rendering the generalised entropy finite and regulator independent [12–15]. $S_{\text{bulk}}$ is the finite piece of the bulk entanglement entropy, and $G_N$ is the renormalised Newton's constant, both of which are renormalisation scheme-dependent.

### 2.1 Bit threads and double holography

First, we take a small but instructive diversion to consider quantum bit threads in doubly holographic models. In double holography we have three equivalent descriptions [10]: a $CFT_d$, which has an (asymptotically) $AdS_{d+1}$ gravitational dual with a bulk matter $CFT_{d+1}$, which in turn has an (asymptotically) $AdS_{d+2}$ gravitational dual. See Fig. 1. The advantage of such models is that the bulk entanglement entropy in $AdS_{d+1}$ can be calculated holographically. We can use the classical Ryu-Takayanagi formula, if we assume the quantum corrections in $AdS_{d+2}$ to be subleading.

In doubly holographic models the quantum bit thread prescription in a sense follows directly from the behaviour of classical bit threads along the boundary branes in the highest dimensional picture. Classical bit threads start at $A$ and can travel through, leave and join the boundary branes in the $AdS_{d+2}$ picture, though they cannot start or end on them [3]. In the $AdS_{d+1}$ picture, these look like bit threads starting and ending at bulk points.

The bulk is classical in the $AdS_{d+2}$ picture, so we use the classical bit thread prescription given in (2). As Fig. 1 illustrates, the bit threads separate into those that stay on the boundary and those that move into the bulk. The one novelty is that what the $G_N$ appearing in the norm

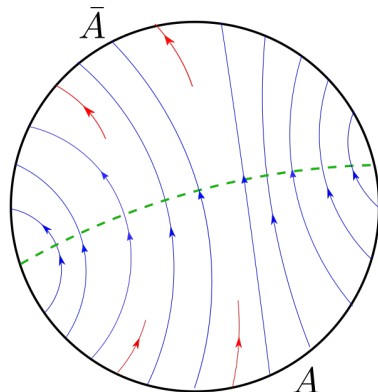

Figure 2: A flux-maximising quantum bit thread flow on a time slice of $\text{AdS}_{d+1}$. Bit threads can start and end at bulk points and some appear to jump across the quantum extremal surface (green dashed line). The number of threads that can appear and disappear is determined by von Neumann entropy of bulk subregions, and in doubly holographic models this is given by RT surface areas in the higher dimensional $\text{AdS}_{d+2}$ geometry (see figure 1).

bound is depends on whether the flow is through the bulk or along the boundary. The flux of bit threads into the bulk from any subregion of the boundary is bounded by the subregion's RT surface.

How does this all look from the $\text{AdS}_{d+1}$ perspective? As Fig. 2 depicts, some bit threads appear to jump over the quantum extremal surface, and double holography gives us an interesting interpretation of these threads as moving into the higher dimensional bulk. The flux density from the boundary of $\text{AdS}_{d+2}$ equals the divergence of $v$ on the $\text{AdS}_{d+1}$ boundary. The bound on flux off subregions of the $\text{AdS}_{d+2}$ boundary translates to

$$\left| \int_\sigma \nabla_\mu v^\mu \right| \leq S_{\text{bulk}}(\sigma), \tag{9}$$

for all subregions $\sigma$ of the $\text{AdS}_{d+1}$ time slice. Here, $\sigma$ need not be a bulk homology region for $A$.

This property of flow configurations in doubly holographic models is very suggestive of how to modify the bit thread prescription to account for quantum corrections. We now turn to consider such modifications. In the discussion section, we will revisit doubly holographic models when we discuss islands from the quantum bit thread perspective.

## 2.2 Warm-up proposals

We do not claim there to be a unique quantum bit thread prescription. That should not surprise us as even the original classical prescription is non-unique; one can trivially generate a family of mathematically equivalent prescriptions by redefining $v^\mu(x) \to f(x)\tilde{v}^\mu(x)$, which superficially look different from the original prescription because $\nabla_\mu \tilde{v}^\mu \neq 0$. The proposals we will discuss are not as trivially equivalent as that, but the point stands.

*Proposal I*

To illustrate what makes some formulations more desirable than others, we consider a short series of iterative improvements, starting from the crudest way to account for quantum corrections in the bit thread formulation, which is to add the entropy of the bulk homology region for the classical RT surface to the final answer by hand

$$S(A) = \max_v \left( \int_A n_\mu v^\mu \right) + S_{\text{bulk}}(\sigma(m_{RT})), \quad \text{subject to } |v| \leq \frac{1}{4G_N}, \quad \text{and } \nabla_\mu v^\mu = 0. \tag{10}$$

This is inaccurate beyond the leading order correction in $G_N$. There are two other reasons why this prescription is lacking. Firstly, it requires detailed foreknowledge about the bulk state and geometry, including where the RT surface will be, to know what $S_{\text{bulk}}$ to add. As in the classical prescription, the threads should have as little foreknowledge about the bulk as possible. Secondly, the quantum correction is added by hand, rather than determined dynamically by the threads themselves, like the area term is.

*Proposal II*

Proposal II is a flow prescription that is equivalent to the FLM surface prescription given in [15]. We can do better than proposal I by modifying divergencelessness condition,

$$S(A) = \max_v \left( \int_A n_\mu v^\mu \right), \quad \text{subject to } |v| \leq 1, \quad \text{and } \nabla_\mu v^\mu(x) = -J(x), \tag{11}$$

with source function $J$ to be determined. We assume that the maximal flow saturates the norm bound on the QES, and then, by the divergence theorem

$$\int_A n_\mu v^\mu = -\int_{\sigma(m)} \nabla_\mu v^\mu + \int_m n_\mu v^\mu, \tag{12}$$

(where the normal vector is outward pointing on $m$, and inward-pointing on $A$) we see that to capture the bulk entropy term and match the QES prescription we require

$$\int_{\sigma(m_{QES})} J(x) = S_{\text{bulk}}(\sigma(m_{QES})). \tag{13}$$

Thus $J$ is some kind of density function for the bulk von Neumann entropy. Such density functions have been considered before, first in [16], and dubbed entanglement contours. In a follow-up paper, we will explore entanglement contours in detail [17].

In previous work, bit thread flux density was interpreted as an entanglement contour [18], which seems different from our proposal, which is that the divergence of the flow field equals an entanglement contour, but the connection is made in doubly holographic models where $\nabla_\mu v^\mu$ in the $\text{AdS}_{d+1}$ picture equals the flux density of bit threads off the boundary in the $\text{AdS}_{d+2}$ picture.

The prescription (11) is an improvement on (10) because the bulk entanglement entropy is accounted for dynamically by the bit threads with only a simple and local modification to the divergencelessness condition. It is still unsatisfactory because it requires foreknowledge of where the Ryu-Takayanagi surface will be, to know what source function $J$ to use, and is only accurate to subleading order in $1/N$.

## 2.3 Quantum bit thread proposal

*Proposal III*

The last prescription we consider requires no foreknowledge and is accurate to all orders in $G_N$. It replaces the divergencelessness condition with a constraint on the flux into all bulk homology regions for $A$.

The prescription is

$$S(A) = \max_v \int_A n_\mu v^\mu, \tag{14}$$

with $v$ subject to the constraints

$$|v| \leq \frac{1}{4G_N} \quad \text{and} \quad \forall(\sigma \in \Omega_A): \quad \left( -\int_\sigma \nabla_\mu v^\mu(x) \leq S_{bulk}(\sigma) \right), \tag{15}$$

and where $\Omega_A$ is the set of all bulk homology regions for $A$:

$$\Omega_A := \{\sigma \subseteq \Sigma : \partial \sigma \supseteq A\}. \tag{16}$$

The prescription needs neither more nor less information about the bulk slice than the QES prescription, as expected. It needs the bulk metric and the renormalised von Neumann entropies of all bulk homology regions for $A$.

Neither of the constraints are regulator-independent, as both the renormalised Newton's constant $G_N$ and the renormalised bulk von Neumann entropy $S_{\text{bulk}}$ in (14) are regulator-dependent. Nevertheless, the constraints together are regulator-independent - one way to argue this is through its equivalence to the QES prescription, which is regulator-independent [13, 19].

How is this prescription equivalent to the QES prescription? Consider an arbitrary surface $m$ homologous to $A$, and its associated bulk region $\sigma(m)$. The constraints and the divergence theorem (12) bound the flux out of $A$

$$\int_A n_\mu v^\mu \leq \frac{\text{Area}(m)}{4G_N} + S_{\text{bulk}}(\sigma(m)), \tag{17}$$

for all $m \sim A$. We can try to saturate this inequality and so turn it into an equality by minimising the right-hand side over surfaces $m$ homologous to $A$, which minimises on $m = m_{QES}$, and maximising the left-hand side with respect to $v$. The inequality is saturated if we assume that there exists a flow field configuration $v$ that:

1. Satisfies the constraints of our prescription (14).

2. Has $v^\mu = n^\mu/4G_N$ on the QES, with $n^\mu$ the unit normal.

3. Saturates the divergence bound when $m = m_{QES}$:

$$-\int_{\sigma(m_{QES})} \nabla_\mu v^\mu = S_{\text{bulk}}(\sigma(m_{QES})). \tag{18}$$

This existence assumption is reasonable, though the precise argument for why is somewhat involved, so to avoid interrupting the flow we will come back to it at the end of the section. With the assumption, (17) becomes

$$\max_v \int_A n_\mu v^\mu = \min_{m \sim A}\left( \frac{\text{Area}(m)}{4G_N} + S_{\text{bulk}}(\sigma(m)) \right), \tag{19}$$

which establishes the equivalence of our quantum bit thread prescription to the QES prescription. We will give a rigorous proof of the equivalence to the QES prescription, which does not rely on existence assumptions, in section 3.

### 2.3.1 Properties

We have modified the classical bit thread proposal by replacing the $\nabla_\mu v^\mu = 0$ condition with a constraint that allows bit threads to start and end at bulk points, but that limit the number of threads that can end in all bulk homology regions. Our prescription is non-local, in that it constrains the flow over codimension-0 (with respect to the bulk time slice) regions; this is not unexpected, and it may be it cannot be improved upon, given the non-local nature of bulk entanglement.

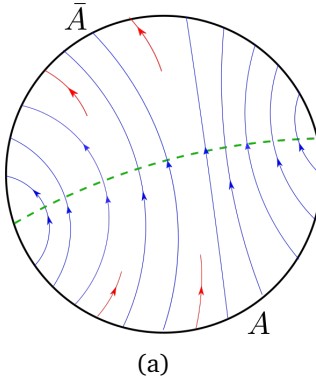
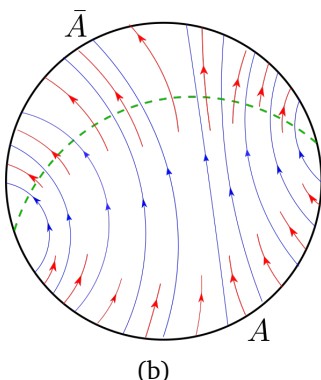

Figure 3: Flux-maximising flows for two different states on the same bulk slice, which illustrates the quantum extremal surface phase transition from the bit thread perspective. The QES, depicted by the green dashed line, is always the bottleneck to the flow. In (a) there is little entanglement across the QES, and the QES is $l_P$ away from the minimal-area classical RT surface. In (b) the bulk entanglement has increased until a new bottleneck appears in the bulk region where the flow is constrained to reappear, which generally will not be close to the classical RT surface.

As expected, the classical bit thread prescription is recovered in the $N \to \infty$ limit. First, we rescale $v$ by $4G_N$, which gives the prescription

$$S(A) = \frac{1}{4G_N} \max_{v} \int_A v \,, \tag{20}$$

with $v$ subject to the constraints

$$|v| \le 1 \text{ and } \forall(\sigma \in \Omega_A): \quad \int_\sigma \nabla_\mu v^\mu(x) \ge -4G_N S_{bulk}(\sigma) \,. \tag{21}$$

There exist states and regions for which $S_{bulk}(\sigma)$ is order $1/G_N$, but with reasonable assumptions and for finite energy states the (renormalised) entropy of any sufficiently small bulk region is $O(1)$, and for these states and regions we have $\lim_{G_N \to 0}(G_N S_{bulk}) = 0$. Next, for the divergence constraint to hold for all $\sigma$ requires $\nabla_\mu v^\mu(x) \ge 0$ for all $x$. If it were not then the constraint would be violated for the $\sigma$ equal to $\Sigma$ minus the neighbourhood of $x$ where $\nabla_\mu v^\mu < 0$. Finally, this constraint $\nabla_\mu v^\mu \ge 0$ can be further tightened, without affecting the optimum, to $\nabla_\mu v^\mu(x) = 0$ - which is the divergencelessness constraint of the original bit thread prescription - because positive sources for $v$ in the bulk cannot possibly increase the flux from $A$.

Another consequence of the constraint: if the global bulk state is pure then when applied to $\sigma = \Sigma \in \Omega_A$ our divergence constraint implies that

$$\int_\Sigma \nabla_\mu v^\mu(x) \ge 0 \,. \tag{22}$$

For pure states, the flux out of bulk slice is thus non-negative; there cannot be more bit threads that end at points in the bulk than start.

Even when the QES and the classical RT surface are not close together (in Planck units), our quantum bit thread prescription (14) is still correct and equivalent to the QES prescription, as we will prove in section 3. Such situations have been of recent interest in the context of the black hole information paradox [20, 21]. The QES and classical RT surfaces are often

perturbatively close to each other, due to the bulk entanglement entropy term having a $G_N$ suppression relative to the area term, but there are cases when the bulk entanglement across a candidate QES is sufficiently large to overcome the relative $G_N$ suppression. As the bulk entanglement across the QES increases, there may come a point where its position jumps discontinuously. Fig. 3 shows what is happening from the quantum bit thread perspective. As the bulk entanglement across the minimal area RT surface increases, more and more threads are allowed to jump across that bottleneck, but the threads eventually become maximally packed in some new region where they are forced to reappear due to the divergence constraint, and a new bottleneck emerges at the quantum extremal surface. As the bulk entanglement increases, there is a continuous change in the flow vector field, in contrast to the discontinuous jump of the extremal surface, and there is still a bottleneck to the flow, but it is no longer the minimal area surface. The QES is *always* the bottleneck to the flow.

The divergence constraint in our quantum bit thread prescription is similar to the property of flows in doubly holographic models we found earlier, which was

$$\forall \sigma \subseteq \Sigma: \quad \left| \int_\sigma \nabla_\mu v^\mu \right| \leq S_{\text{bulk}}(\sigma). \tag{23}$$

Doubly holographic models are a subset of all holographic models, so, for consistency, their flows, which satisfy (23), must also satisfy the constraints of our quantum bit thread prescription given in (14), which they do.

The property (23) may or may not be too tight a constraint to impose in non-doubly-holographic models. It is stronger than the constraint in (14) in two ways: it applies to all bulk subregions, not just bulk homology regions for $A$, and the absolute value is taken. Taking the absolute value is certainly too strong; $S_{bulk}$ is a renormalised entropy, it can be negative, and the constraint (23) can not be satisfied for any $v$ when it is.

### 2.3.2 Quantum bit threads near the QES

Now we come back to justify the assumption that was key to establishing equivalence of prescriptions, that there exists a flow field $v$ that satisfies the conditions given in (18), one of which was that it saturates the norm bound with $v^\mu = n^\mu/4G_N$ on the QES. There is cause to question this assumption because in the classical prescription (with $\nabla_\mu v^\mu = 0$) it is *not* possible to saturate the norm bound $|v| \leq 1/4G_N$ on any surface homologous to $A$ other than the minimal area one without violating the norm bound elsewhere. In the quantum prescription, however, due to the non-zero divergence of $v$, threads can start and end anywhere in the bulk, and this makes it possible for the norm bound to be saturated on a non-minimal area surface, such as the QES, without the norm bound being violated immediately off the surface.

Since we assumed that the bit threads of a flux maximising flow are maximally packed on the QES, the most sensible place to check whether the norm bound will be violated is in the neighbourhood of the QES. From the definition of the QES, shape deformations of its generalised entropy vanish to first order, which implies that

$$0 = \frac{\delta}{\delta m(x)} \left( \frac{\text{Area}(m)}{4G_N} + S_{\text{bulk}}(\sigma(m)) \right) = \frac{K(x)}{4G_N} + \frac{\delta}{\delta m(x)} S_{\text{bulk}}(\sigma(m)), \tag{24}$$

on the QES, with $K$ the trace of the extrinsic curvature. Suppose now also that the divergence bound is saturated not only for $m = m_{QES}$ as in (18), but for first order shape deformations away from the QES too, then

$$\nabla_\mu v^\mu(x) = -\frac{\delta}{\delta m(x)} S_{\text{bulk}}(\sigma(m)), \tag{25}$$

on the QES. By assumption $|v| = 1/4G_N$ on the QES and what we now check is whether our supposition (25) is sufficient to ensure that the norm bound $|v| \leq 1/4G_N$ is not violated at first order in directions perpendicular to the QES; this requires $v^\mu \partial_\mu |v| = 0$. This equation was called the 'linear obstruction equation' in [7] and was shown to be equivalent to

$$\nabla_\mu v^\mu(x) = \frac{K(x)}{4G_N}, \tag{26}$$

on any surface on which $v^\mu = n^\mu/4G_N$ (which for us is the QES). Now if (26) were incompatible with the constraints of our prescription then we would be in trouble, but (24) and (25) do together imply (26), which shows that the supposition (25) (which is within our prescription's constraints) is sufficient.

## 3 Proof of equivalence to the QES prescription

At the end of the last section, we did a local analysis in the neighbourhood of the QES and found strong evidence for the existence of flux-maximising flows that make the quantum bit thread and QES prescriptions equivalent. In this section, with a completely different approach, we *prove* the equivalence.[3] We will use tools from convex optimisation, which were also used by the authors of [3] to prove the equivalence of the classical Ryu-Takayanagi and bit thread prescriptions.

The key tool is the application of Lagrangian duality, the basic idea of which is to

1. Take a constrained optimisation problem, the 'primal'.

2. Introduce Lagrange multiplier terms to enforce the constraints.

3. Optimise over the original variables, leaving us with a new optimisation problem called the 'dual'.

For us, the primal problem is the quantum bit thread prescription and the target dual problem is the quantum extremal surface prescription.

The steps to finding the Lagrangian dual of a constrained optimisation problem which we've enumerated are mathematically straightforward, and we do not need to go into the general theory of Lagrangian duality and convex optimisation, but readers who would like to learn more about the mathematical background can read the review in section 2 of [3] and the references therein.

The only non-trivial result from convex optimisation we need is Slater's condition, which is a sufficient condition for strong duality to hold. Strong duality means that the optima of the primal and dual problems are equal. Without strong duality, the primal and dual optimisation problems are not equivalent, which to this section's purpose would be fatal.

### 3.1 From the quantum bit thread prescription to a dual optimisation problem

Lagrangian dualisation begins with a constrained optimisation problem, the primal. Our bit thread proposal meets the definition of a special class of optimisation problems, called concave optimisation problems, because both the objective function we are maximising over

$$\int_A v, \tag{27}$$

---

[3]To a physicist's level of rigour.

and the constraints we are imposing

$$\frac{1}{4G_N} - |v| \geq 0 \,, \tag{28}$$

and

$$\forall(\sigma \in \Omega_A): -\int_\sigma \nabla_\mu v^\mu \leq S_{\text{bulk}}(\sigma) \,, \tag{29}$$

are concave functions of $v$. Recall that $\Omega_A$ is the set of bulk homology regions for $A$, and $S_{bulk}$ is the renormalised bulk von Neumann entropy of the reduced state. In our simplified notation, $\int_A v$ is short for $\int_A \sqrt{h} n_\mu v^\mu$ with $n$ the inward pointing unit normal to the bulk Cauchy slice, and we suppress the argument of $v$ and the measure on $\Sigma$.

Slater's condition is satisfied by the quantum bit thread prescription. This is important because if Slater's condition is satisfied for a convex or concave optimisation problem then strong duality holds. Slater's condition requires there to exist a strictly feasible point in the domain of the primal problem. A strictly feasible point is a point that satisfies the constraints, including *strictly* satisfying all the non-linear constraints. For us (28) is the only non-linear constraint that needs to be strictly satisfied, and $v = 0$ is a strictly feasible point, so strong duality holds. This establishes that our quantum bit thread proposal is equivalent to whatever optimisation problems we can turn it into using our Lagrangian dualisation procedure.[4]

The next step of the Lagrangian dualisation procedure is to add Lagrange multiplier terms for each constraint. The divergence constraint given by (29) applies to every element of $\Omega_A$, and the norm bound (28) applies at every point in the bulk time slice $\Sigma$. Adding Langrange multiplier terms for these constraints gives

$$\sup_v \inf_{\mu,\phi} \left[ \int_A v + \int_\Sigma \phi(x)\left(\frac{1}{4G_N} - |v|\right) + \int_{\Omega_A} d\mu(\sigma)\left(\left(\int_\Sigma \chi(\sigma,x)\nabla_\mu v^\mu\right) + S_{\text{bulk}}(\sigma)\right) \right], \tag{30}$$

where $\mu$ is a (non-negative) measure on $\Omega_A$, $\phi$ a non-negative[5] scalar function on $\Sigma$, and $\chi(\sigma,x)$ the characteristic function for $\sigma \subseteq \Sigma$ defined by

$$\chi(\sigma,x) := \begin{cases} 1, & \text{for } x \in \sigma \,, \\ 0, & \text{for } x \in \Sigma \backslash \sigma \,. \end{cases} \tag{31}$$

Minimising with respect to the Lagrange multipliers $\mu$ and $\phi$ would return us to the primal problem. Instead, we maximise with respect to the original variable $v$, after integrating the $\nabla_\mu v^\mu$ term by parts, which gives[6]

$$\inf_{\mu,\phi} \sup_v \left[ \left(1 - \int_{\Omega_A} d\mu(\sigma)\right)\int_A v + \int_\Sigma \left(\phi(x)\left(\frac{1}{4G_N} - |v|\right) - v^\mu \partial_\mu \left(\int_{\Omega_A} d\mu(\sigma)\chi(\sigma,x)\right)\right) + \int_{\Omega_A} d\mu(\sigma) S_{\text{bulk}}(\sigma) \right], \tag{32}$$

The supremum of this with respect to $v$ is $+\infty$, unless both

$$\int_{\Omega_A} d\mu(\sigma) = 1 \,, \tag{33}$$

---

[4]As in [3] we assume that what is true about convex optimisation problems with a finite number of constraints is also true when there are an infinite number of constraints.

[5]Lagrange multipliers usually enforce equality constraints, while we have inequality constraints, so $\mu$ and $\phi$ are a generalisation called Karush–Kuhn–Tucker multipliers, which is also why $\mu$ and $\phi$ need to be non-negative.

[6]We have assumed that the orders of integration can be reversed.

and

$$\phi(x) \geq \left| \partial_\mu \int_{\Omega_A} d\mu(\sigma) \chi(\sigma, x) \right|. \tag{34}$$

Since we are minimising with respect $\mu$ and $\phi$, we certainly want to avoid the region of the domain of the optimisation problem (32) where either (33) or (34) are not satisfied and the optimum is $+\infty$. With that in mind, (32) is equivalent to

$$\inf_{\mu, \phi} \left[ \frac{1}{4G_N} \int_\Sigma \phi(x) + \int_{\Omega_A} d\mu(\sigma) S_{\text{bulk}}(\sigma) \right], \tag{35}$$

as long as we impose (33) and (34) as constraints. The constraint (33) tells us that $\mu$ is a probability measure on $\Omega_A$. The minimisation of (35) with respect to $\phi$ is trivial; it saturates the inequality (34). This leaves us with a Lagrangian dual to our quantum bit thread prescription, which is to find

$$\inf_\mu \left[ \frac{1}{4G_N} \int_\Sigma \left| \partial_\mu \int_{\Omega_A} d\mu(\sigma) \chi(\sigma, x) \right| + \int_{\Omega_A} d\mu(\sigma) S_{\text{bulk}}(\sigma) \right], \quad \text{subject to} \quad \int_{\Omega_A} d\mu(\sigma) = 1. \tag{36}$$

What remains to show is that this dual problem is equivalent to the QES prescription. We have two ways of showing this.

## 3.2 From the dual optimisation problem to the QES prescription

**Argument 1**

Our first argument starts by noting that if in (36) we could take the $\int_{\Omega_A} d\mu(\sigma)$ outside of the norm, then our optimisation problem would be

$$\inf_\mu \left[ \int_{\Omega_A} d\mu(\sigma) \left( \frac{1}{4G_N} \int_\Sigma |\partial_\mu \chi(\sigma, x)| + S_{\text{bulk}}(\sigma) \right) \right] = \inf_\mu \left[ \int_{\Omega_A} d\mu(\sigma) \left( \frac{\text{Area}(m)}{4G_N} + S_{\text{bulk}}(\sigma) \right) \right]. \tag{37}$$

This is a minimisation of generalised entropy over probability measures on the set of bulk homology regions for $A$, which is equivalent to the QES prescription and what we wanted. The first equality uses that the normal derivative of the characteristic function $\chi(\sigma, x)$ is a surface delta function on $m$, with $\partial \sigma = m \cup A$.

What stops us from taking $\int_{\Omega_A} d\mu(\sigma)$ outside the norm? The first term in (36) can be thought of as the norm of the integral of such surface delta functions over those $m$ on which $\mu(\sigma)$ has support. $\mu$ can have support on surfaces whose contributions to $\int_\Sigma \left| \partial_\mu \int_{\Omega_A} d\mu(\sigma) \chi(\sigma, x) \right|$ partially cancel against each other inside the norm. For this cancellation to happen for a pair of bulk regions $\sigma_1$ and $\sigma_2$, with boundaries $\partial \sigma_1 = m_1 \cup A$ and $\partial \sigma_2 = m_2 \cup A$, the surfaces $m_1$ and $m_2$ must have some overlap with oppositely oriented surface normals, an example of which is shown in Fig. 4. To spell out the issue, for the $m_1$ and $m_2$ shown in Fig. 4 the following is not true:

$$\int_\Sigma |\partial_\mu \chi(\sigma_1, x) + \partial_\mu \chi(\sigma_2, x)| = \int_\Sigma \left( |\partial_\mu \chi(\sigma_1, x)| + |\partial_\mu \chi(\sigma_2, x)| \right). \tag{38}$$

Now, suppose we are given a measure $\mu$ that has support on a pair of bulk regions that are problematic in the sense we have been discussing; they do not satisfy (38). Since we are minimising over probability measures on $\Omega_A$, it would be sufficient to show that we can define a new measure $\mu'$ from $\mu$, which does not increase the value of the objective function we are minimising over, and does not have support on both $\sigma_1$ and $\sigma_2$.

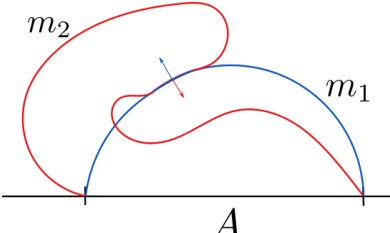
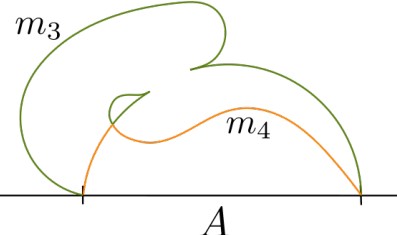

Figure 4: $m_1$ and $m_2$ are two surfaces homologous to $A$ whose intersection is such that their contributions to the first term in (36) partially cancel. By cutting and pasting we can define new surfaces $m_3$ and $m_4$ with lower total bulk entropy and whose contributions to the first term in (37) do not cancel.

Take such a probability measure $\mu$ with support on $\sigma_1$ and $\sigma_2$, and assume without loss of generality that $d\mu(\sigma_1) \leq d\mu(\sigma_2)$. Define new bulk homology regions that are also in $\Omega_A$

$$
\begin{aligned}
\sigma_3 &:= \sigma_1 \cup \sigma_2, \\
\sigma_4 &:= \sigma_1 \cap \sigma_2,
\end{aligned}
\tag{39}
$$

and a new probability measure $\mu'$ which is identical to $\mu$ except we move support from $\sigma_1$ and $\sigma_2$ to $\sigma_3$ and $\sigma_4$:

$$
\begin{aligned}
d\mu'(\sigma_1) &= 0, \\
d\mu'(\sigma_2) &= d\mu(\sigma_2) - d\mu(\sigma_1), \\
d\mu'(\sigma_3) &= d\mu(\sigma_3) + d\mu(\sigma_1), \\
d\mu'(\sigma_4) &= d\mu(\sigma_4) + d\mu(\sigma_1).
\end{aligned}
\tag{40}
$$

Changing $\mu \mapsto \mu'$ cannot increase the objective function in (36); the first term is unchanged, and the entropy term cannot increase because of the strong subadditivity of bulk entropies

$$
\begin{aligned}
S_{\text{bulk}}(\sigma_1) + S_{\text{bulk}}(\sigma_2) &\geq S_{\text{bulk}}(\sigma_1 \cap \sigma_2) + S_{\text{bulk}}(\sigma_1 \cup \sigma_2) \\
&= S_{\text{bulk}}(\sigma_3) + S_{\text{bulk}}(\sigma_4).
\end{aligned}
\tag{41}
$$

For an arbitrary measure $\mu$ we imagine repeating this process, tracing a path through the space of probability measures, until we are left with a measure with support only on surfaces which do not cancel one another inside the norm of (36), and whose evaluation in the objective function of the Lagrangian dual can not be greater than that of the original measure. This establishes that (36) and (37) are equivalent, and so our quantum bit thread prescription and the QES prescription are equivalent.

**Argument 2**[7]

Strong subadditivity of the bulk entropies is also key to the second way of arguing equivalence of our Lagrangian dual problem to the QES prescription. We start from an entropy inequality that follows from the strong subadditivity relation being repeatedly applied to the von Neumann entropies of $N$ regions [22]:

$$
\sum_{i=1}^{N} S(X_i) \geq S(\cup_i X_i) + S(\cup_{\{ij\}} X_i \cap X_j) + \ldots + S(\cap_i X_i).
\tag{42}
$$

We apply this relation to an arbitrary set of $N$ bulk homology regions $\sigma_i$ (with as usual $\partial \sigma_i = m_i \cup A$). The argument of the $n$th term on the right-hand side of (42) is the bulk

---

[7]We credit Matt Headrick with the basic idea behind this argument.

region $r(n/N)$, which is the union of all points which are in at least a fraction $n/N$ of those bulk regions $\sigma_i$,

$$r(n/N) := \{x \in \Sigma : \psi(x) \geq n/N\}, \tag{43}$$

with $\psi(x)$ the fraction of $\sigma_i$ that contain the point $x$,

$$\psi(x) := \frac{1}{N} \sum_{i=1}^{N} \chi(\sigma_i, x). \tag{44}$$

With these definitions (42) becomes

$$\sum_{i=1}^{N} S_{\text{bulk}}(\sigma_i)) \geq \sum_{i=1}^{N} S_{\text{bulk}}(r(i/N)), \tag{45}$$

which in the $N \to \infty$ limit (after dividing both sides by $N$) becomes

$$\int_{\Omega_A} d\mu(\sigma) S_{\text{bulk}}(\sigma) \geq \int_0^1 dp\, S_{\text{bulk}}(r(p)), \tag{46}$$

with $\mu$ again an arbitrary probability measure on $\Omega_A$. This is the first relation we need to prove equivalence of (36) to the QES prescription.

In the $N \to \infty$ limit, $\psi(x)$ as defined in (44) becomes the expression that appears in the first term of (36):

$$\psi(x) = \int_{\Omega_A} d\mu(\sigma) \chi(\sigma, x). \tag{47}$$

From the fact that $\mu$ is a probability measure, $\psi$ satisfies $0 \leq \psi(x) \leq 1$ and $\psi(x)|_{\partial\Sigma} = \chi(A, x)$, where $\chi(A, x)$ is the characteristic function of $A \subseteq \partial\Sigma$, whose domain is $\partial\Sigma$ and equals 1 for $x \in A$ and 0 for $x \in \partial\Sigma\backslash A$.

From the codimension-0 subregions we defined earlier

$$r(p) := \{x \in \Sigma : \psi(x) \geq p\}, \tag{48}$$

where $p$ is now a continuous variable in $[0, 1]$, we define the associated codimension-1 level sets $m(p) := \partial r(p)\backslash A$. This allows us to rewrite the first term in the objective function of (36) as the average area of the level sets $m(p)$ (recalling that the normal derivative of a characteristic function is a surface delta function):

$$\int_\Sigma \left| \partial_\mu \int_{\Omega_A} d\mu(\sigma) \chi(\sigma, x) \right| = \int_\Sigma |\partial_\mu \psi(x)| = \int_0^1 dp\, \text{Area}(m(p)). \tag{49}$$

This is the second relation we need, which in combination with (46), allows us to write the objective function of our Lagrangian dual optimisation problem as

$$\int_0^1 dp \left( \frac{\text{Area}(m(p))}{4G_N} + S_{\text{bulk}}(r(p)) \right). \tag{50}$$

$r(p)$ with its associated level set $m(p)$ is determined by $\mu$ through (47) and (48). Minimising this function with respect to $\mu$ is equivalent to the QES prescription; the function is minimised when $\mu$ only has support on the entanglement wedge time slice, when

$$\psi(x) = \int_{\Omega_A} d\mu(\sigma) \chi(\sigma, x) = \chi(\sigma(m_{QES}), x). \tag{51}$$

This completes our proof that the quantum bit thread prescription is equivalent to the quantum extremal surface prescription.

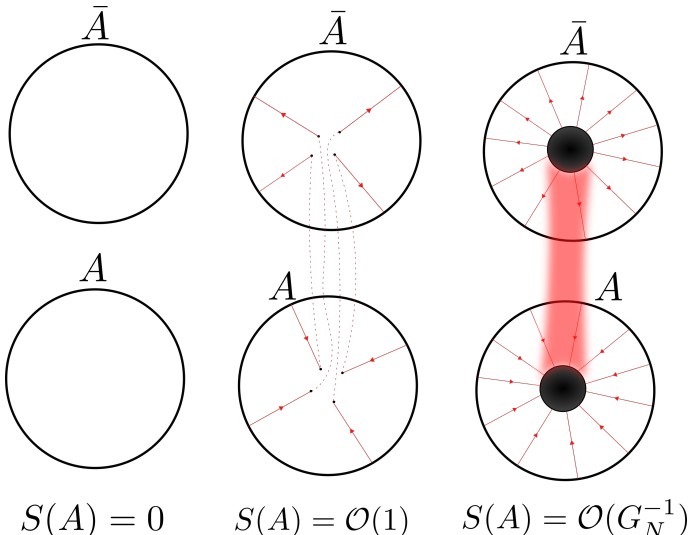

$$S(A) = 0 \qquad S(A) = \mathcal{O}(1) \qquad S(A) = \mathcal{O}(G_N^{-1})$$

Figure 5: It from qubit threads? The figure depicts three pairs of holographic CFTs with differing degrees of entanglement between them, showing how quantum bit threads are a mechanism (at least pictorially) for bulk spacetime emergence. The red lines depict a flux-maximising flow configuration, the dashed red lines connect entangled bulk degrees of freedom, and the fuzzy red region represents the macroscopic wormhole.

## 4 Discussion and future directions

Quantum bit threads give an interesting perspective on ER = EPR [23] and the emergence of spacetime from entanglement. Consider the set-up shown in Fig. 5, whose three subfigures depicts unentangled, weakly entangled, and strongly entangled pairs of holographic CFTs. Our prescription tells us that the number of threads that can jump from one bulk to the other is limited by the bulk entanglement entropy. In the first pair of CFTs, the state is unentangled between them, so no threads can jump. In the second pair, the CFTs are weakly entangled, and we suppose that there are a few EPR pairs shared between the two bulk spacetimes which allows an order $\mathcal{O}(1)$ number of threads to jump across. Rather than thinking of the quantum bit threads as jumping discontinuously from one bulk to the other, one could alternatively imagine that each quantum bit thread, loosely speaking, travels between bulks through its own wormhole of Planck scale cross-section. In the last pair of CFTs we keep adding EPR pairs until we eventually form an entangled pair of black holes joined by a classical wormhole. If we take this microscopic wormhole idea seriously, then the natural interpretation of what has happened is that the $\mathcal{O}(G_N^{-1})$ microscopic wormholes have coalesced into a single macroscopic wormhole. This speculative interpretation, that bit threads are in some sense Planck scale wormholes, and that bulk spacetimes are built from them, is a direction it would be interesting to explore further.

Fig. 6 shows a flux-maximising flow configuration in the island phase of a doubly holographic set up.[8] To capture the Area$(\partial \mathcal{I})$ term of the island formula, in the AdS$_{d+2}$ picture some bit threads in the flux-maximising flow must rejoin the boundary in the island region and travel through $\partial \mathcal{I}$. From the quantum bit thread perspective in AdS$_{d+1}$, islands are regions

---

[8]We expect that the comments we make here about islands and quantum bit threads are true for more than just doubly holographic models. We use these models because the behaviour of flux-maximising quantum bit thread flows follows immediately from the behaviour of classical bit threads in the highest dimensional picture, which are easier to work with.

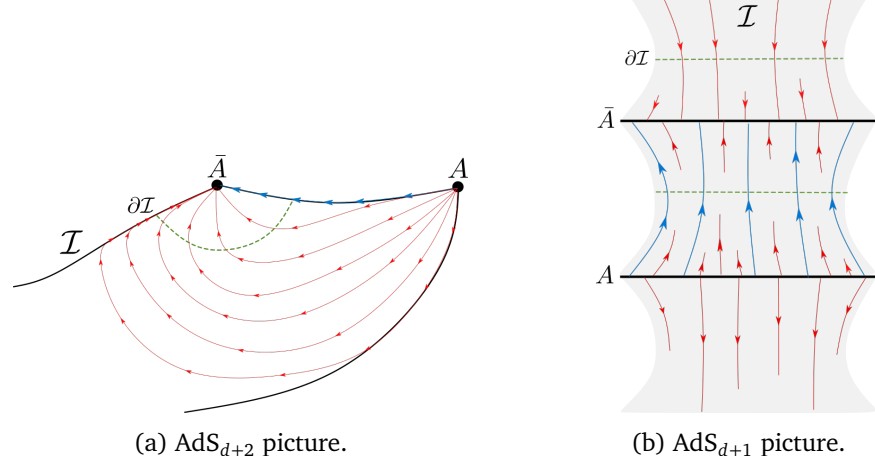

(a) AdS$_{d+2}$ picture.      (b) AdS$_{d+1}$ picture.

Figure 6: Quantum bit threads in the island phase of a doubly holographic model. $A$ and $\bar{A}$ are a pair of CFT$_d$'s, and the lines with arrows are representative of a flux-maximising flow from $A$ to $\bar{A}$. From the AdS$_{d+1}$ perspective, the quantum bit threads start from $A$, disappear at some bulk point, reappear elsewhere, then end on $\bar{A}$. Some reach $\bar{A}$ by reappearing in the island region $\mathcal{I}$, whose boundary $\partial\mathcal{I}$ is an emergent bottleneck for quantum bit threads in the island phase.

of the bulk where, due to the entanglement structure of the bulk state, the bit threads of flux-maximising flows are constrained to reappear; so many that a new bottleneck emerges. The island phase transition is not discontinuous for bit threads like it is for the surfaces involved. The island is disjoint from the rest of $A$'s entanglement wedge, but they are connected by the quantum bit threads that jump from one to the other; another manifestation of ER=EPR. In some sense islands are an emergent phenomenon that materialise when too many quantum bit threads try to reappear in a bulk region.

We found a prescription for bit threads that is accurate to all orders in $1/N$, to leading order in the 't Hooft coupling $\lambda$, and valid for time reflection symmetric states. A significant goal of the bit thread program is to find a prescription that is as widely applicable as current surface-based holographic entanglement entropy constructions. There are covariant bit thread prescriptions, valid at leading order in $1/N$ and for non-time reflection symmetric states [24], so it is natural to look for a covariant quantum bit thread prescription. This is important because both time dependence and quantum corrections are necessary to apply holographic entanglement entropy to evaporating black holes. Regarding finite $\lambda$ corrections, classical bit threads have been generalised to bulk actions with higher curvature gravitational terms, such as Gauss-Bonnet, see [7], and there are no obvious obstacles to directly combining finite $\lambda$ and finite $N$ corrections to bit threads.

We can ask whether there are other quantum bit thread prescriptions than the ones we have discussed. We do not know whether it is possible to have a prescription that modifies the divergence constraint in a local way, say to $\nabla_\mu v^\mu(x) = J(x)$, without needing an oracle to tell us where the entangling surface will be as in proposal II.

We have proven equivalence of one particular proposal to the QES prescription, but not uniqueness, nor, as we have mentioned, should we expect uniqueness. One step in this direction is to ask the extent to which one could loosen or tighten constraints in the prescription we have given. Our stance has been to place constraints on $v$ that are sufficiently strong so that the flux through $A$ does not exceed $S(A)$, but no stronger than that. One such tightening of constraints would be to turn the divergence inequality constraint into an equality, but this is too strong; it would require (25) to be satisfied at each $x$ for all surfaces $m$ that pass through

that point, which is generally impossible.

## Acknowledgments

I would like to thank Matt Headrick and Qiang Wen for useful discussions and comments on an earlier draft. I would also like to thank Gurbir Arora, Ben Freivogel, Harsha Hampapura and Juan Pedraza for useful discussions. This research was supported by the Stichting Nederlandse Wetenschappelijk Onderzoek Instituten (NWO-I) through 'Scanning New Horizons', DoE grant DE-SC0009987 and the Simons Foundation.

## A   Nesting and entropy inequalities

We want to prove that quantum bit threads satisfy the nesting property. This is because there are flow-based proofs of the subadditivity and strong subadditivity entropy inequalities which require the nesting property to hold [2]. Since the Araki-Lieb inequality and positivity of von Neumann entropy follow by subadditivity, a proof that quantum bit threads satisfy the nesting property is an indirect proof of an important set of entropy inequalities.

The nesting property for flows says that for any pair of boundary regions $A$ and $B$ there exists a flow $v$ which simultaneously maximises the flux through both $A$ and $AB$.[9] We want to show that there exists a $v$, which satisfies the quantum bit thread prescription's constraints, such that

$$\int_{AB} v = S(AB) \quad \text{and} \quad \int_{A} v = S(A), \tag{A.1}$$

where $S(A)$ is the von Neumann entropy of boundary region $A$, to all orders in $1/N$, as calculated by either the QES or quantum bit thread prescriptions.

To prove that the nesting property holds, it is sufficient to show that

$$\sup_{v} \left( \int_{A} v + \int_{AB} v \right), \tag{A.2}$$

equals $S(A) + S(AB)$.

The flow $v$ is subject to the norm bound

$$|v| \le \frac{1}{4G_N}, \tag{A.3}$$

and a constraint that makes $v$ a valid quantum bit thread flow for both $A$ and $AB$

$$\forall (\sigma \in \Omega_A \cup \Omega_{AB}): \quad -\int_{\sigma} \nabla_{\mu} v^{\mu} \le S_{bulk}(\sigma), \tag{A.4}$$

$\Omega_A$ is the set of bulk homology regions for boundary subregion $A$:

$$\Omega_A := \{\sigma \subseteq \Sigma : \partial \sigma \supseteq A\}, \tag{A.5}$$

and $\Omega_{AB}$ the corresponding set of bulk homology regions for $AB$.

The supremum (A.2) cannot be greater than $S(A) + S(AB)$ because the supremum of the sum cannot be greater than the sum of the suprema, so if we can also show that it cannot be

---

[9]$AB$ is short-hand for $A \cup B$.

less than $S(A) + S(AB)$ then it must be equal. This is a proof strategy that we are adapting from [3].

To (A.2) we add Karush-Kuhn-Tucker multiplier terms with non-negative multipliers for the inequality constraints:

$$
\sup_v \inf_{\phi,\mu,\mu'} \left[ \int_A v + \int_{AB} v + \int_\Sigma \phi(x)\left(\frac{1}{4G_N} - |v|\right) + \int_{\Omega_A} d\mu(\sigma)\left(\left(\int_\Sigma \chi(\sigma,x)\nabla\cdot v(x)\right) + S_{bulk}(\sigma)\right) \right.
$$
$$
\left. + \int_{\Omega_{AB}} d\mu'(\sigma)\left(\left(\int_\Sigma \chi(\sigma,x)\nabla\cdot v(x)\right) + S_{bulk}(\sigma)\right) \right], \tag{A.6}
$$

and then we integrate by parts the terms with derivatives of $v$

$$
\sup_v \inf_{\phi,\mu,\mu'} \left[ \left(2 - \int_{\Omega_A} d\mu - \int_{\Omega_{AB}} d\mu'\right)\int_A v + \left(1 - \int_{\Omega_{AB}} d\mu'\right)\int_B v - \int_\Sigma v\cdot\partial\psi(x) + \int_\Sigma \phi(x)\left(\frac{1}{4G_N} - |v|\right) \right.
$$
$$
\left. + \int_{\Omega_A} d\mu(\sigma) S_{bulk}(\sigma) + \int_{\Omega_{AB}} d\mu'(\sigma) S_{bulk}(\sigma) \right]. \tag{A.7}
$$

We have used that, by its definition, $\chi(\sigma,x) = 1$ for $x \in A$ and 0 for $x \in (\partial\Sigma\backslash A)$ for all $\sigma \in \Omega_A$. The $\mu'$ in the above equations should not be confused with that defined in section 3.2; at this stage $\mu$ and $\mu'$ are independent measures on different sets of bulk homology regions. We have also introduced a new quantity $\psi$ defined by

$$
\psi(x) := \int_{\Omega_A} d\mu(\sigma)\chi(\sigma,x) + \int_{\Omega_{AB}} d\mu'(\sigma)\chi(\sigma,x). \tag{A.8}
$$

We are free to switch the order in which the supremum and infimum are taken in (A.7) because Slater's conditions are satisfied and so strong duality holds - the constraint (A.4) is linear in $v$, and the non-linear constraint (A.3) has the strictly feasible point $v = 0$.

Let us then switch the order: take the supremum over $v$, and then the infimum over $\mu$, $\mu'$ and $\phi$. The supremum over $v$ on $\partial\Sigma$ is infinite unless the coefficients of the boundary $v$ flux terms in (A.7) are zero, so a finite infimum of the supremum must have $\mu$ and $\mu'$ which are probability measures on $\Omega_A$ and $\Omega_{AB}$ respectively:

$$
\int_{\Omega_A} d\mu = 1, \qquad \int_{\Omega_{AB}} d\mu' = 1. \tag{A.9}
$$

This in turn implies that $0 \le \psi(x) \le 2$ for $x \in \Sigma$, and that on the boundary of $\Sigma$ we have $\psi(x) = 2$ on $A$, $\psi(x) = 1$ on $B$, and $\psi(x) = 0$ on $\partial\Sigma\backslash(AB)$.

Just as in (34) the supremum over $v$ in the interior of the bulk slice is infinite unless $\phi(x) \ge |\partial\psi(x)|$, and taking the infimum with respect to $\phi$ saturates this inequality. This leaves us with

$$
\inf_{\mu,\mu'} \left[ \frac{\int_\Sigma |\partial\psi(x)|}{4G_N} + \int_{\Omega_A} d\mu\, S_{bulk}(\sigma) + \int_{\Omega_{AB}} d\mu'\, S_{bulk}(\sigma) \right]. \tag{A.10}
$$

Next, we use a simple extension of the result (46) which is

$$
\int_{\Omega_A} d\mu S_{bulk}(\sigma) + \int_{\Omega_{AB}} d\mu' S_{bulk}(\sigma) \ge \int_0^2 dp\, S_{bulk}(r(p)). \tag{A.11}
$$

We have defined

$$
r(p) := \{x \in \Sigma \,|\, \psi(x) \ge p\}, \tag{A.12}
$$

which has the properties

$$
\begin{aligned}
r(p \leq 0) &= \Sigma, \\
r(0 < p \leq 1) \cap \partial\Sigma &= A \cup B, \\
r(1 < p \leq 2) \cap \partial\Sigma &= A, \\
r(2 < p) &= \varnothing.
\end{aligned}
\tag{A.13}
$$

Equation (A.11) is proven the same way as (46): start from (42), except with $2N$ bulk regions split evenly between $\Omega_A$ and $\Omega_{AB}$.

We also need

$$
\int_\Sigma |\partial\psi(x)| = \int dp \, \text{Area}(m(p)),
\tag{A.14}
$$

where $m(p) := (\partial r(p) \backslash \partial\Sigma)$. Note that $m(p) = \varnothing$ for $p \leq 0$ and $p > 2$, so the right hand side of (A.14) only gets a contribution from $0 < p \leq 2$.

Plugging (A.14) and (A.11) into (A.10) we arrive at the result that the supremum (A.2) is lower bounded by

$$
\inf_{\mu,\mu'} \left[ \int_0^2 dp \left( \frac{\text{Area}(m(p))}{4G_N} + S_{bulk}(r(p)) \right) \right].
\tag{A.15}
$$

Equation (A.15) is equal to the QES result for $S(AB) + S(A)$ because $m(p)$ is homologous to $AB$ for $0 < p \leq 1$, and to $A$ for $1 < p \leq 2$; the infimum has $\mu$ equal to a delta-function measure which picks out the bulk homology region for the QES homologous to $A$, and similarly for $\mu'$ with respect to $AB$.

This completes the proof of the nesting property.

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
