# Peer review of "Quantum bit threads"

_SciPost Physics, doi:SciPost Phys. 14, 097 (2023)_

## Round 2 · Referee Report · Anonymous (Referee 1) · 2022-10-31

Strengths

  1. Provides flow reformulation of QES prescription for computing holographic entanglement entropy using quantum bit threads.
  2. Relates quantum bit threads to `double holography'.
  3. Comments on relation of quantum bit threads to the emergence of islands.
  4. Well written, clear and concise.

Weaknesses

  1. Relation between quantum bit threads and double holography could be expanded.
  2. Relation between quantum bit threads and islands could be expanded.

Report

Overall the article is well written and fills an important gap in the literature on holographic entanglement entropy and bit threads. The article meets the standards of Scipost and therefore recommend this article for publication, upon a few minor requested changes.

Requested changes

  1. Respond to my two questions regarding braneworld holography and bit threads (attached file).
  2. Update references.

Attachment

  • validity: high
  • significance: high
  • originality: high
  • clarity: high
  • formatting: perfect
  • grammar: perfect

Author:  Andrew Rolph  on 2022-11-28  [id 3083]

(in reply to Report 1 on 2022-10-31)

I would like to thank the referee for the insightful questions and comments. I will answer the questions in the attached report one by one.

  • Do these higher curvature corrections play a role in determining the bound on the flux, from the intermediate brane perspective?

Good question. The short answer is yes. In figure 1, the highest dimensional description, the bit threads maximise their flux from the codim-2 region $A$ to $\bar A$, and a flux-maximising flow flows through both the codim-0 bulk, and the IR brane (the codim-1 lower boundary in the figure). The more threads that can flow through the IR brane, the more threads that can and will leave the upper boundary in a flux-maximising flow. Brane actions can have higher curvature terms in the gravitational sector, and the bit thread prescription in the prescence of such terms has a modified norm bound, as described in 1807.04294. So, in a given doubly holographic model, the divergence constraint in the intermediate picture (figure 2) will be depend partly and indirectly on the brane action. That said, the main result of the subsection, the constraint in eqn (2.2), is model-independent. To be clear, what the right hand side of (2.2) evaluates to depends on the region $\sigma$, the model, and the state, but the actual relation (2.2) is independent of all those things.

  • That is, can we think of these corrections being the reason for the modified flux bound, owed to a change in the change in norm bound due to the presence of higher derivative terms?

Whether or not the IR brane has higher curvature terms does not affect the relation (2.2), only what the right-hand side evaluates to. It may be useful to say that stringy and quantum corrections to holographic entanglement entropy are as distinct in flow-based prescriptions as they are in surface-based prescriptions.

  • How is the argument modified when the brane as a flat or dS brane geometry? Or what about when there is a black hole on the brane?

The geometry and state on the IR brane will affect right-hand side of (2.2), i.e. bulk entropies in the intermediate picture, but again won't affect the actual relation.

  • Can a Lorentzian analog of quanutm bit threads (namely, the modified divergenceless condition) be used to propose a modification of the complexity=volume conjecture?

Trivially, modifying the divergenceless condition in the `Lorentzian' bit thread proposal will affect the minimal flux, and for a generic modification, the resulting prescription will not be equivalent to complexity=volume. Such modifications may still result in a prescription that satisfies the properties one wants of complexity. It would be interesting which such modifications do or do not, to pick a property of complexity, and have a non-negative minimal flux.

  • Lastly, can the author comment on whether the quantum bit thread proposal provides an easier route to derive monogamy of mutual information?

Given the difficulty of proving monogamy of mutual information with classical bit threads, at leading order in $1/N$, I would not expect it to be easier with $1/N$ corrections included.

  • Does one need to assume the bulk entanglement entropy obeys MMI?

The short answer is yes. At finite $N$, the boundary entanglement entropy satisfies MMI if the bulk entanglement entropy satisfies MMI (see arXiv:2108.07280). It is a sufficient condition, though, to my knowledge, unknown if it is necessary. The condition is independent of whether the boundary EE is calculated using the flow or surface-based holographic prescriptions, so if it is necessary for the surface formulation then it is necessary for the flow formulation, and vice versa.

---

## Round 2 · Referee Report · Anonymous (Referee 2) · 2022-11-14

Strengths

1- Generalizes the 'bit thread' prescription for computing entanglement entropy in holographic settings to include $1/N$ corrections (analogous to the QES prescription). 2- Provides intuition for its interpretation from models of double holography. 3- Sheds light on the phenomenon of entanglement islands.

Weaknesses

1- Limited to static or time-symmetric cases. Does not apply to the case of islands emerging in the context of black hole evaporation. 2- Requires knowledge of all possible $S_\text{bulk}(\sigma)$ (analogous to the QES prescription) so it is not clear how to use the formula in practice. 3- Besides the interpretation, it is not clear what the advantage of the new prescription may be (e.g. if it can be used to prove any new property of holographic entanglement).

Report

The paper "Quantum bit threads" develops a novel prescription to compute holographic entanglement entropy to all orders in $1/N$, which generalizes the flow prescription developed by Freedman and Headrick a few years ago. The paper is well-written and well-organized. The analyses and derivations are solid, and as far as I am concerned, are physically reasonable. The results of this paper deepen our understanding of holographic entanglement and, therefore, would be a worthy addition to the literature.

Before I recommend the paper for publication, I have a few minor questions I would like the author to address:

1) The divergence constraint (1.6 - right) only involves $S_\text{bulk}$, which is in principle UV divergent. It is only when added with the area term that such UV divergences may be canceled, so it is not entirely clear to me that this constraint gives something non-trivial? On the other hand, if one takes $S_\text{bulk}$ to be the finite piece I would naively think the constraint is scheme dependent.

2) Related to my first question: I was under the impression that the divergences from the area term and the bulk entropy cancel out only for the correct extremal surface but not in general. Is there an argument to see if this cancelation may be possible for a general bulk surface homologous to the region of interest?

3) Is it assumed that the Hilbert space can be arbitrarily factorized in the bulk? If so, it should be explicitly stated (note this may not be possible in a theory containing gravity).

4) Intuitively, what would happen with the constraints (1.6) and with quantum bit threads in situations where the homology constraint is non-trivial? For example, for a large region in a spherical black hole background? (in these situations the standard RT surface is disconnected and contains the black hole horizon as one of its components)

5) Regarding Fig 1, it is not so clear to me that there could be an unambiguous separation between the two types of threads. Classical bit threads normally enter deep into the bulk. Is there a concrete example to illustrate this separation?

6) The second proposal seems relevant (and sufficient) to describe the FLM prescription (as opposed to the QES prescription). I think this should be stated in the paper.

Once these points are addressed, I would be happy to revisit my recommendation to the journal.

Requested changes

1- Please address my questions (1-6) in the main body of the report.

  • validity: high
  • significance: good
  • originality: good
  • clarity: high
  • formatting: excellent
  • grammar: perfect

Author:  Andrew Rolph  on 2022-11-28  [id 3082]

(in reply to Report 2 on 2022-11-14)
Category:
answer to question

I would like to thank the referee for the insightful questions and comments and will answer the questions one by one.

1) It is correct that the renormalised bulk entanglement entropy is scheme dependent, and that this makes bit thread divergence constraint in eq (1.6) scheme dependent. So too are the renormalised Newton's constant in (1.6) and the norm-bound constraint. Both constraints are scheme dependent, but the full quantum bit thread proposal is scheme independent; one way to argue is through the proof of equivalence to the QES formula, which is terms of the scheme independent generalised entropy. The divergence constraint (1.6) can also be rewritten in terms of generalised entropy.

2) To my knowledge, the cancellation is not special to extremal surfaces. For an arbitrary bulk region, the leading UV divergence of its entanglement entropy is proportional to the boundary area, and, at least for the states explored in arXiv:1302.1878, those divergences exactly cancel the counterterms in the gravitational effective action.

3) Yes, just as in the quantum extremal surface prescription, I neglect possible but poorly understood contributions of metric fluctuations to bulk entanglement entropy and issues of factorisation of the bulk Hilbert space.

4) Consider the black hole example described in the question, where the boundary of the bulk homology region is disconnected and can include the horizon, and suppose that the bulk entropy of that region is dominated by the black hole's entropy. The quantum bit thread divergence constraint then says that the number of bit threads that may end in that region is upper bounded approximately by the black hole's entropy. This is what we want for a one-sided mixed state AdS black hole: a flux-maximising flow configuration that calculates the entropy of the entire boundary has bit threads which start on the boundary and end in the bulk, and the number of threads is given by the black hole entropy.

5) The separation is into flow lines which either are or are not purely along the top boundary in fig 1. From the intermediate picture perspective, shown in fig 2, where we project onto the top boundary, the separation maps into flow lines that either do or do not have non-zero sources. It is not clear to me what the confusion is.

6) It is correct that the second proposal is equivalent to the FLM prescription, and I agree that this should be stated in the paper.

---

## Round 3 · Referee Report · Anonymous (Referee 1) · 2022-12-21

Strengths

  1. Provides flow reformulation of QES prescription for computing holographic entanglement entropy using quantum bit threads.
  2. Relates quantum bit threads to `double holography'.
  3. Comments on relation of quantum bit threads to the emergence of islands.
  4. Well written, clear and concise.

Report

This journal's acceptance criteria has been met. Therefore I recommend this article for publication.

Requested changes

None.

---

## Round 3 · Referee Report · Anonymous (Referee 2) · 2023-1-3

Report

The author has considered my comments and implemented the necessary corrections in the manuscript. I'm happy with the current version and recommend the article for publication.

---

## Round 3 · Author Response

The referees asked some questions and suggested some changes. In my resubmission, I've made the changes I said I would in the responses to the referees.

---

## Round 3 · List of Changes

To address referee 2's comment #3, I've added "Regarding bulk entanglement entropy, we neglect graviton fluctuations and potential issues of bulk
Hilbert space factorisation, as in the FLM and QES proposals" to footnote 2.

To address referee 2's comment #6, I've added "Proposal II is a flow prescription that is equivalent to the FLM surface prescription" to page 6.

To address referee 1's requested change #2, I updated the references to include https://arxiv.org/abs/2208.10507

---

## Editorial Decision

published